# Veda: Scalable Video Diffusion via Distilled Sparse Attention

**Shihao Han** [1 2 *]  **Hao Yang** [1 †]  **Xiaofeng Mei** [1]  **Xinting Hu** [3]  **Yi Jiang** [1]  **Xiaojuan Qi** [2]

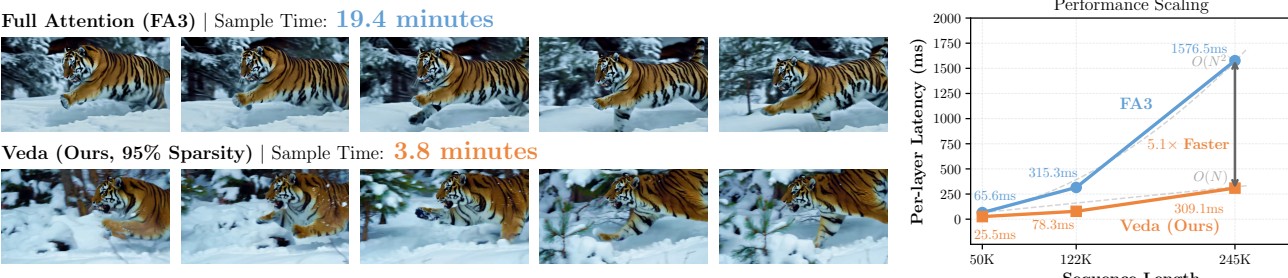

**Figure 1.** **Veda achieves a 5.1× end-to-end GPU speedup for high-resolution, long-video generation.** Evaluated on Waver-T2V-12B (720P / 241 frames), Veda at 95% sparsity reduces the sample time from 19.4 to 3.8 minutes without compromising visual quality.

## Abstract

Scaling Diffusion Transformers to generate high-resolution, long videos is constrained by the quadratic cost of self-attention, and existing sparse attention methods degrade under high sparsity. We show empirically that generation quality is determined not by the sparsity ratio itself, but by how well the sparse mask aligns with the tile-wise geometry of full attention. Based on this insight, we propose Veda, a distilled sparse attention framework that formulates tile selection as an explicit reconstruction problem from full attention. Veda integrates statistics-aware tile scoring with head-aware tiling to reduce estimation error and structural mismatch, enabling aggressive sparsity. A hardware-efficient tile-skipping kernel converts theoretical sparsity into practical wall-clock speedups. Experiments on large video diffusion models, including Waver and Wan2.1, demonstrate substantial acceleration with no noticeable degradation in generation quality. To generate 720P 10-second videos on Waver-T2V-12B, Veda achieves a 5.1× end-to-end speedup and a 10.5× self-attention speedup, reducing attention overhead from 92% to 50%. Notably, the gains in-

crease with sequence length, indicating that Veda scales favorably with spatiotemporal resolution across models.

## 1. Introduction

Diffusion Transformers (DiTs) (Peebles & Xie, 2023) have emerged as the leading architecture for high-fidelity video synthesis (Lin et al., 2024; StepVideoTeam, 2025; Wan-Team, 2025; Team et al., 2025; Yang et al., 2025; Hunyuan-Team, 2025). Nevertheless, their scalability is bottlenecked by the quadratic computational and memory complexity of self-attention (Vaswani et al., 2017) with respect to the spatiotemporal token sequence length. Sparse attention offers a natural path toward alleviating this bottleneck by restricting attention computation to a subset of token pairs. In practice, however, sparse attention must be realized at the granularity of tiles rather than individual tokens to align with modern GPU hardware, which performs attention through block-wise matrix multiplications (Markidis et al., 2018; Zhang et al., 2025c). This reformulates sparse attention as a ***tile selection*** problem: tokens are grouped into tiles, and each query tile attends to only a small number of key tiles, producing a tile-wise mask that can be directly materialized by a tile-skipping kernel for efficient execution.

However, translating sparsity into wall-clock acceleration often comes at the cost of generation quality. To preserve spatiotemporal structure, existing approaches construct tile masks with two main paradigms. Methods such as SVG (Xi et al., 2025) and STA (Zhang et al., 2025c) leverage inductive biases of pretrained models to define candidate spa-

[1]ByteDance Inc. [2]The University of Hong Kong [3]University of Science and Technology of China [*]Work done during internship at ByteDance. [†]Project Lead. Correspondence to: Xinting Hu <xinting@ustc.edu.cn>, Xiaojuan Qi <xjqi@hku.hk>.

*Proceedings of the 43rd International Conference on Machine Learning*, Seoul, South Korea. PMLR 306, 2026. Copyright 2026 by the author(s).

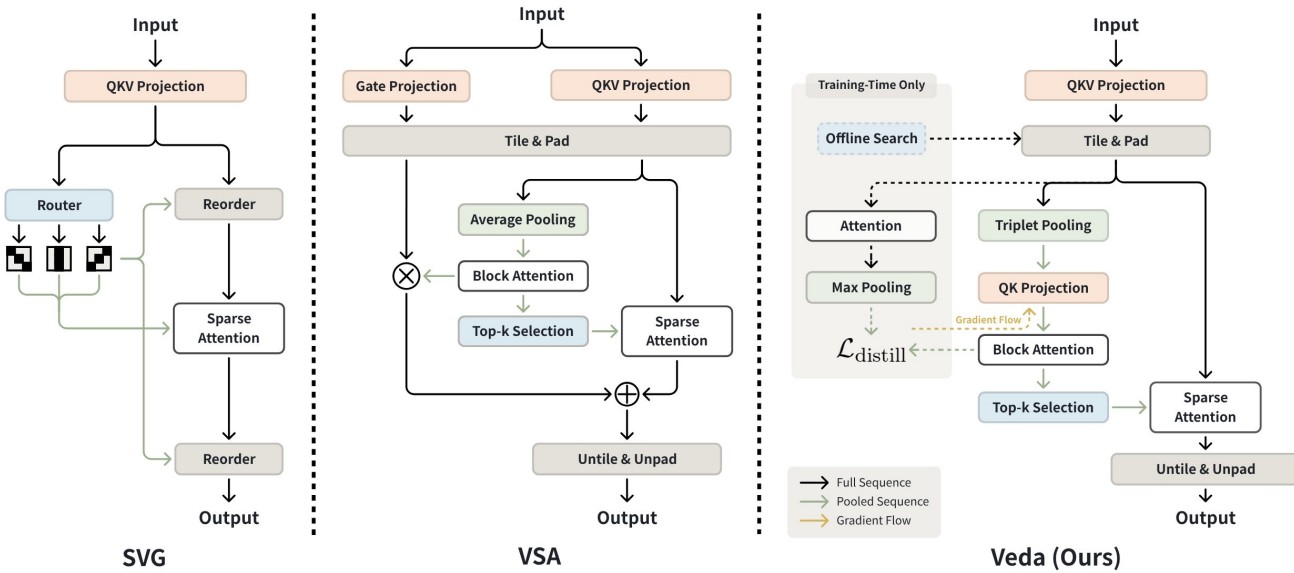

*Figure 2.* **Architectural comparison of sparse attention methods.** We compare the proposed Veda (ours) with representative methods that use (i) static sparse patterns (SVG (Xi et al., 2025)) and (ii) dynamically learned sparse masks (VSA (Zhang et al., 2025b)). Solid lines indicate the inference data flow, while dashed lines denote auxiliary components used only during training.

tiotemporal masks and employ online or offline search strategies to select the most effective ***pre-defined*** patterns (see Fig. 2, SVG). While these methods are simple and hardware-friendly, they lack adaptability to the highly dynamic and head-specific attention structures learned by DiTs.

In contrast, VMOBA (Wu et al., 2025) and VSA (Zhang et al., 2025b) adopt dynamic tile selection by estimating tile importance from compressed representations, such as pooled features or low-rank approximations, and ranking tiles to synthesize ***dynamic*** tile-selection masks. Although more flexible (see Fig. 2, VSA), these approaches suffer from inaccurate estimation: tile importance is learned only implicitly through the diffusion objective, without explicit supervision to preserve the structural geometry of full attention. Moreover, their reliance on coarse statistics, such as mean pooling, fails to capture salient signal peaks within tiles, causing the estimated tile scores to misrepresent true query-key correlations. As a result, both paradigms exhibit pronounced quality degradation (Fig. 3) when sparsity is pushed beyond moderate levels, producing structured artifacts including spatial warping, water-ripple patterns, and temporal flickering.

To understand what limits sparse attention in video diffusion, we conduct an empirical study on the relationship between sparsity and generation quality (see Fig. 4). We construct an "Oracle" sparse mask by max-pooling the full-attention matrix into a tile-level score map and retaining the highest-response tiles, thereby inducing a reference per-query ranking of key tiles. Models using this Oracle mask preserve generation quality even under very high sparsity,

demonstrating that sparsity itself is not the primary cause of degradation (Fig. 4). Instead, we find that mask quality, specifically, the degree to which the sparse mask aligns with the tile-wise structure of full attention, dominates performance.

Consistent with this observation, existing methods fail for two complementary reasons: methods with pre-defined sparse patterns suffer from structural mismatch with head-specific attention geometry, while dynamic methods incur estimation error due to implicit supervision and insufficient tile statistics. Furthermore, we observe substantial heterogeneity across attention heads and diffusion timesteps, indicating that a single, uniform tiling strategy is inherently suboptimal. This suggests that sparse attention must control two coupled factors: how tokens are grouped into tile, and which tiles are selected.

Guided by these findings, we propose Veda (Fig. 2, Veda), a distilled sparse attention framework that preserves the tile-wise structure and ranking of full attention even under extreme sparsity. Unlike prior dynamic methods in which tile selection emerges implicitly from the diffusion objective, Veda learns tile selection as an explicit target. Specifically, Veda employs a lightweight estimator to distill tile-level attention scores from a full-attention backbone. To reduce estimation error, we enrich tile representations with extrema-aware statistics that preserve salient peaks beyond simple mean pooling. To further mitigate structural mismatch, Veda introduces a Head-Aware Tiling strategy: under a fixed hardware tile budget, heads that primarily capture local spatial interactions and those that

model long-range temporal dependencies are assigned distinct spatiotemporal factorizations of tiles. Together, these components enable aggressive pruning of attention tiles while maintaining alignment with the intrinsic structure of full attention. Finally, to translate theoretical FLOPs reduction into practical end-to-end acceleration, we implement a tile-skipping sparse attention kernel. This kernel materializes the predicted sparsity through efficient tile skipping and reaches approximately $80\%$ of the MFU achieved by FlashAttention-3 (Shah et al., 2024) while executing sparse attention, ensuring that sparsity yields tangible latency gains rather than overhead-dominated speedups.

We conduct extensive experiments on the Waver and Wan2.1 video diffusion models and consistently observe substantial acceleration with no noticeable degradation in generation quality. Across both backbones, Veda maintains stable visual fidelity and temporal coherence even at high sparsity, indicating that the speed gains do not come at the cost of structural artifacts. On Waver-T2V-12B generating 720P 10-second videos, Veda achieves a $5.1\times$ end-to-end speedup and a $10.5\times$ self-attention speedup, reducing attention overhead from 92% to 50%. On Wan2.1-T2V-14B 720P 5-second, Veda further delivers a $2.63\times$ end-to-end speedup with a $7.08\times$ acceleration in self-attention. Notably, the speedup increases with sequence length, demonstrating that Veda scales favorably with spatiotemporal resolution across models (Fig. 8).

## 2. Related Work

**Sparse Attention in Language Modeling** Sparse attention was first developed and stress-tested at scale in language modeling (Touvron et al., 2023; QwenTeam, 2025; DeepSeekTeam, 2025a). Early works introduced static sparsity constraints like Sliding Window Attention (Hassani et al., 2023; Fu et al., 2024) and StreamingLLM (Xiao et al., 2023) that restrict attention to local neighborhoods or specific sinks to maintain efficiency. More recent approaches have evolved toward dynamic sparsity, utilizing either heuristic approximations (*e.g.*, MMInference (Li et al., 2025), MoBA (Lu et al., 2025), NSA (Yuan et al., 2025)) or predictive mechanisms that estimate significant attention tiles (e.g., SeerAttention (Gao et al., 2025), DSA (DeepSeek-Team, 2025b)). Our approach is closely related to this predictive paradigm. However, direct transfer to video generation is non-trivial: tightly coupled spatiotemporal dependencies can amplify small sparsity errors into structured distortions and temporal flicker, motivating video-specific sparsity designs.

**Sparsity Mechanisms for Video Synthesis** Sparsifying the attention matrix in Video DiTs offers a pathway to efficient long-video generation. Works such as SVG (Xi et al., 2025), Sparse-vDiT (Chen et al., 2025), VORTA (Sun et al., 2025)

and STA (Zhang et al., 2025c) rely on static sparsity patterns, which are either manually defined or derived from offline statistics. To produce content-adaptive sparsity at runtime, methods like VSA (Zhang et al., 2025b) and VMOBA (Wu et al., 2025) rely on the backbone network to implicitly learn the optimal sparse structure alongside the video generation task. Despite wall-clock gains, the learned selection can drift away from the tile structure implicitly relied upon by the pretrained full-attention model, leading to visible artifacts under high sparsity. This motivates our key stance: learning where to attend benefits from explicit objectives that preserve the tile-wise structure of full attention.

**Efficiency for Video DiTs Beyond Sparse Attention** Beyond sparse attention, research on accelerating Video DiTs generally follows complementary paradigms. Methods such as PAB (Zhao et al., 2025) and TeaCache (Fan et al., 2025) reuse intermediate activations across denoising steps, but the locality assumption can break in few-step regimes where large timestep jumps induce substantial feature disparity. In parallel, sampling distillation methods such as CausVid (Yin et al., 2025) and rCM (Zheng et al., 2025) reduce the number of model evaluations by compressing the diffusion trajectory, yet the per-step spatial cost remains quadratic ($O(N^2)$). Our sparse attention framework is orthogonal to both directions and can be combined with them to maximize end-to-end acceleration.

## 3. What Matters for a Good Sparse Model?

### 3.1. Preliminary

Modern video diffusion DiTs (Peebles & Xie, 2023) model global dependencies over spatiotemporal tokens, yet attention quickly becomes the primary computation and memory bottleneck at high resolution and long duration. Specifically, a video latent of shape $(T, H, W)$ is flattened into a token sequence of length $N = THW$. For a single attention head, let $\mathbf{Q}, \mathbf{K}, \mathbf{V} \in \mathbb{R}^{N \times d}$ denote the query, key, and value matrices. Attention (Vaswani et al., 2017) is computed as

$$\mathbf{A} = \text{Softmax}\left(\frac{\mathbf{Q}\mathbf{K}^\top}{\sqrt{d}}\right), \qquad \mathbf{O} = \mathbf{A}\mathbf{V}, \qquad (1)$$

which incurs $O(N^2)$ computation and memory complexity. To mitigate the quadratic cost, sparse attention introduces a mask to restrict computation to a subset of token pairs. Following VSA (Zhang et al., 2025b), this can be written as $\mathbf{A} = \text{Softmax}(\mathbf{Q}\mathbf{K}^\top/\sqrt{d} + \mathbf{M})$, where full attention corresponds to $\mathbf{M} = \mathbf{0}$, and sparsity is induced by setting most entries of $\mathbf{M}$ to $-\infty$.

Modern hardware accelerators (e.g., GPUs with Tensor Cores) are optimized for tile-level data movement and computation. Accordingly, we formulate sparse attention at the tile granularity. Concretely, $N$ tokens are grouped into $N_T$

tiles of size $B$, yielding tiled tensors $\widetilde{\mathbf{Q}}, \widetilde{\mathbf{K}}, \widetilde{\mathbf{V}} \in \mathbb{R}^{N_T \times B \times d}$. $\widetilde{\mathbf{M}}_{ij} \in \{0, 1\}$ denotes a binary tile-selection mask for the $(i, j)$ tile pair; selected tile pairs are evaluated by the sparse kernel, while unselected pairs are equivalently assigned additive mask value $-\infty$ in the softmax. Under a fixed Top-$k$ budget, each query tile $\widetilde{\mathbf{Q}}_i$ attends to $k$ key tiles and masks out the rest, yielding a tile mask $\widetilde{\mathbf{M}}$ that satisfies

$$\left| \{ j \mid \widetilde{\mathbf{M}}_{ij} = 1 \} \right| = k, \qquad \forall i \in \{1, \dots, N_T\}.$$

The sparse attention for query tile $\widetilde{\mathbf{Q}}_i$ is computed as:

$$\hat{\mathbf{K}}_i = \text{Concat}_{j : \widetilde{\mathbf{M}}_{ij}=1}(\widetilde{\mathbf{K}}_j), \quad \hat{\mathbf{V}}_i = \text{Concat}_{j : \widetilde{\mathbf{M}}_{ij}=1}(\widetilde{\mathbf{V}}_j),$$

$$\mathbf{O}_i = \text{Softmax}\left( \frac{\widetilde{\mathbf{Q}}_i \hat{\mathbf{K}}_i^\top}{\sqrt{d}} \right) \hat{\mathbf{V}}_i. \tag{2}$$

This enables a tile-skipping kernel to bypass the masked tiles for practical wall-clock speedups.

### 3.2. Empirical Study

We first consider two common sparse attention designs that follow the pipeline in 3.1: (i) *Static-pattern* methods that instantiate a tile-structured mask $\widetilde{\mathbf{M}}$ from pre-defined spatiotemporal layouts; and (ii) *Dynamic* methods that construct tile representations via tile pooling and predict tile importance for per-query Top-$k$ selection.

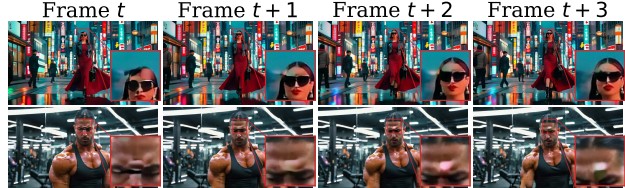

*Figure 3.* **Sparsity-Induced Artifacts.** At 90% sparsity, samples generated with average pooling exhibit "water-ripple" distortions and stochastic noise, which are distinct from standard hallucinations.[1]

**Observation 1: structural artifacts emerge at high sparsity.** At high sparsity ($\geq$ 90%), both static and dynamic baselines exhibit a distinct class of *structural artifacts* that are different from semantic hallucinations of the base model. Typical failures include water-ripple patterns, local geometric distortions, and temporal flickering across frames (Fig. 3).

**Observation 2: mask quality dominates sparsity ratio.** To disentangle the effect of the sparsity ratio from mask

---

[1]Human faces are used because video artifacts are subtle in static images, whereas facial distortions are immediately perceptible to humans.

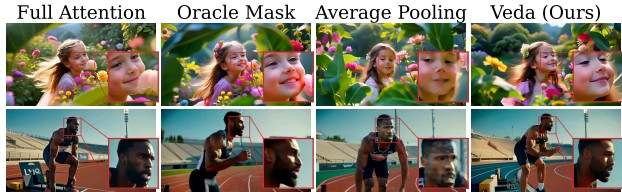

*Figure 4.* **Impact of Mask Precision.** At a fixed sparsity level (90%), generation quality is governed by the accuracy of tile selection. The *Oracle Mask*, selected from full-attention Top-$k$ scores, preserves high-fidelity structure and serves as an upper bound. In contrast, the *Average Pooling* mask used in VSA (Zhang et al., 2025b) misestimates tile importance and causes severe artifacts. Veda distills the oracle selection into an efficient sparse mask, recovering oracle-like quality under the same budget.

quality, we perform a controlled intervention at a fixed sparsity level (e.g., 90%). Using the "optimal" mask derived from full attention scores yields substantially better generations than a pooling-based mask under the same sparsity budget (Fig. 4). This gap indicates that the bottleneck is not the sparsity ratio itself, but whether the tile mask preserves the tile-wise structure of full attention.

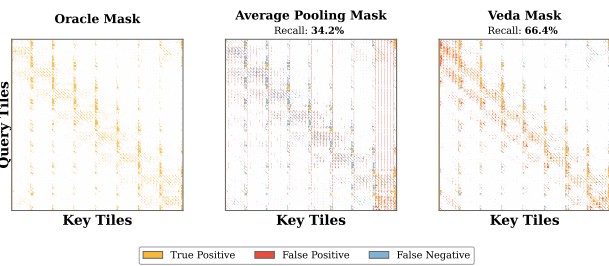

*Figure 5.* **Tile recall analysis against full attention oracle.** Visualization of tile-level selection agreement with the full attention oracle mask (Top-$k$ tiles by aggregated attention scores). Orange indicates true positives (correctly selected tiles), red denotes false positives (incorrectly selected), and blue shows false negatives (missed tiles).

**Metric: tile recall w.r.t. full attention.** To quantify tile-wise alignment, we define a simple recall metric against full attention. For each query tile $i$, let $\widetilde{\mathbf{M}}_i^{\text{fu}}$ be the mask of Top-$k$ key tiles under full attention (computed by aggregating token-level scores into tile scores), and $\widetilde{\mathbf{M}}_i^{\text{sp}}$ be the mask of selected Top-$k$ tiles of a sparse method. Define the selected key-tile sets as $\mathcal{S}_i^{\text{fu}} = \{j \mid \widetilde{\mathbf{M}}_{ij}^{\text{fu}} = 1\}$ and $\mathcal{S}_i^{\text{sp}} = \{j \mid \widetilde{\mathbf{M}}_{ij}^{\text{sp}} = 1\}$. We report *tile recall*:

$$\text{Recall@}k = \frac{1}{N_T} \sum_{i=1}^{N_T} \frac{|\mathcal{S}_i^{\text{sp}} \cap \mathcal{S}_i^{\text{fu}}|}{k}. \tag{3}$$

Across sparsity levels, higher tile recall correlates strongly with fewer structural artifacts, making it a reliable indicator of stability under extreme sparsity (Fig. 5).

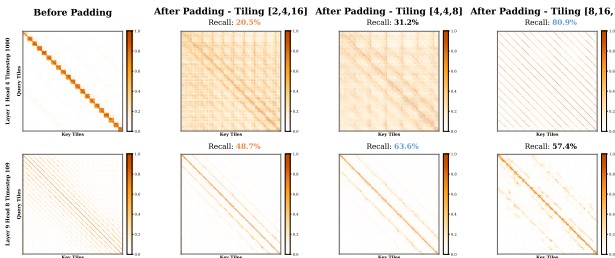

*Figure 6.* **Diversity of Attention Patterns.** Distinct attention heads exhibit varying spatial and temporal footprints that evolve across diffusion timesteps, indicating the need for head-specific adaptation.

**Observation 3: head-wise heterogeneity challenges uniform tile designs.** Finally, inspecting attention patterns reveals pronounced head-wise diversity across layers and diffusion timesteps (Fig. 6). A single, uniform tiling strategy can under-aggregate critical correlations for some heads while over-aggregating others, which degrades tile recall under high sparsity and motivates head-aware tile designs.

## 4. Veda

We introduce Veda, a sparse attention framework designed to distill the intrinsic structure and ranking of full attention. As a lightweight, architecturally modular component, Veda actively aligns tile selection with the full-attention backbone even under extreme sparsity. The overall architecture is shown in Fig. 2, and this section is organized as follows. In Sec. 4.1, Veda casts tile selection as an explicit alignment objective: a statistics-aware score estimator is supervised to recover full-attention query-key tile scores. Sec. 4.2 addresses head-wise heterogeneity with Head-Aware Tiling, assigning head-dependent spatiotemporal tile configurations to reduce structural mismatch. Finally, Sec. 4.3 presents a hardware-efficient tile-skipping kernel that materializes the predicted masks into practical end-to-end acceleration.

### 4.1. Distilled Tile Scoring for Sparse Attention

Veda optimizes a binary tile mask $\widetilde{\mathbf{M}}$ under a fixed Top-$k$ budget to preserve the tile-wise structure and ranking induced by full attention. Rather than relying on the diffusion objective to shape sparsity implicitly, we cast tile selection as an explicit alignment problem with supervision derived from the full-attention backbone.

**Target: Full-Attention Score Construction.** To guide the sparse selection, we first derive a reference tile ranking from the full-attention backbone. Given the query and key features $\mathbf{Q}, \mathbf{K} \in \mathbb{R}^{N \times d}$, the full attention map is $\mathbf{A}^* = \text{Softmax}(\mathbf{Q}\mathbf{K}^\top / \sqrt{d})$. To map this token-level density to tile-level importance, we apply a max-pooling operation

over the corresponding query-key tile regions:

$$\mathbf{S}_{ij}^{\text{tgt}} = \max_{(u,v) \in \text{Tile}(i,j)} \mathbf{A}_{uv}^*. \tag{4}$$

We opt for max-pooling rather than averaging because attention distributions are typically sparse and peaky; averaging tends to dilute salient high-frequency signals with background noise, whereas max-pooling preserves the existence of critical dependencies.

**Statistic-Aware Estimator.** We employ a lightweight estimator $\mathcal{E}_\phi$ to reconstruct the target scores using compressed tile representations. First, for each query/key tile, we construct a TripPool descriptor by concatenating Avg, Max, Min triplet statistics:

$$\text{TripPool}[\cdot] = \text{Avg}[\cdot] \oplus \text{Max}[\cdot] \oplus \text{Min}[\cdot], \tag{5}$$

where $\oplus$ denotes the concatenation. To capture heterogeneous dependency patterns, the estimator employs head-specific MLP projections $\phi_q$ and $\phi_k$. For a given attention head, these projections map the statistics into a shared latent space. The predicted score $\mathbf{S}_{ij}^{\text{pred}}$ between query tile $\widetilde{\mathbf{Q}}_i$ and key tile $\widetilde{\mathbf{K}}_j$ is then computed as:

$$\mathbf{S}_{ij}^{\text{pred}} = \frac{\phi_q(\text{TripPool}[\widetilde{\mathbf{Q}}_i]) \cdot \phi_k(\text{TripPool}[\widetilde{\mathbf{K}}_j])^\top}{\sqrt{d'}}, \tag{6}$$

where $d'$ is the estimator latent dimension. Note that while we omit the head index for brevity, distinct projection weights are learned for each head.

**Optimization.** We align the predicted tile-score distribution to the full-attention reference via a row-wise distillation objective. Concretely, we apply row-wise normalization on $\mathbf{S}^{\text{tgt}}$ and Softmax on $\mathbf{S}^{\text{pred}}$ over key tiles to obtain per-query attention weights $\mathbf{A}^{\text{tgt}}$ and $\mathbf{A}^{\text{pred}}$, and minimize

$$\mathcal{L}_{\text{distill}} = \mathcal{D}_{\text{KL}}\left(\mathbf{A}^{\text{tgt}} \| \mathbf{A}^{\text{pred}}\right), \tag{7}$$

averaged over query tiles and heads.

By applying per-query Top-$k$ selection on $\mathbf{S}^{\text{tgt}}$ and $\mathbf{S}^{\text{pred}}$, we retain the $k$ highest-scoring key tiles for each query tile and mask out the rest, constructing the oracle and predicted binary tile masks $\widetilde{\mathbf{M}}^*$ and $\widetilde{\mathbf{M}}$. The resulting mask is materialized by a tile-skipping sparse attention kernel to compute sparse attention outputs. The backbone is trained under this sparse execution with the standard diffusion denoising objective $\mathcal{L}_{\text{diff}}$, while $\mathcal{L}_{\text{distill}}$ provides explicit supervision to improve tile-score estimation and the induced Top-$k$ selection.

Crucially, to ensure the estimator actively adapts to the backbone without perturbing the pre-trained generative manifold, we apply a stop-gradient (sg) operation on the backbone

features feeding into the estimator. This decouples mask learning from feature learning, ensuring stable convergence. In our experiments, we observed that allowing the gradient to propagate back into the base model leads to a noticeable degradation in generation quality. This empirical finding indicates that implicitly coupling sparse mask learning with the diffusion objective is detrimental; forcing the generative backbone itself to learn how to perform sparse attention fundamentally disrupts its pre-trained representation capabilities.

### 4.2. Head-aware Tiling Search

Attention heads in Video DiTs exhibit pronounced heterogeneity in their spatiotemporal dependency patterns, so a single uniform tiling can be suboptimal under high sparsity. Veda therefore assigns a per-layer-per-head tiling configuration $\pi_{l,h} = (p_t, p_h, p_w)$: For each head $h$ at layer $l$, $\pi_{l,h}$ defines how the $N$ tokens are grouped into $N_T$ tiles over temporal, spatial-height, and spatial-width dimensions. To remain hardware-efficient, we restrict $\pi$ to factorizations of the hardware tile size $B$ (e.g., $B=128$), and the configuration set $\Omega$ is:

$$\Omega = \{(p_t, p_h, p_w) \in \mathbb{N}^3 \mid p_t p_h p_w = B\}, \quad (8)$$

We select $\pi_{l,h}$ offline on a calibration set by directly minimizing the approximation error of full-attention output $\mathbf{O}^{\text{fu}}$ after applying sparse attention. For each candidate $\pi \in \Omega$ and calibration sample $x \sim \mathcal{D}_{\text{cal}}$, we first tile the head-specific tokens according to $\pi$ and derive a reference Top-$k$ mask $\widetilde{\mathbf{M}}^*(x; \pi)$ from the full-attention map, following the score pooling and per-query Top-$k$ selection in Sec. 4.1. We then apply $\widetilde{\mathbf{M}}^*(x; \pi)$ to execute tile-sparse attention and obtain the corresponding output $\mathbf{O}^{\text{sp}}_{l,h}(x; \pi)$ after mapping tokens back to the original order, while $\mathbf{O}^{\text{fu}}_{l,h}(x)$ denotes the full-attention output of the same head. The optimal configuration is chosen as

$$\pi^*_{l,h} = \arg\min_{\pi \in \Omega} \mathbb{E}_{x \sim \mathcal{D}_{\text{cal}}} \left[ \left\| \mathbf{O}^{\text{fu}}_{l,h}(x) - \mathbf{O}^{\text{sp}}_{l,h}(x; \pi) \right\|^2_F \right],$$
$$(9)$$

which favors tilings that preserve the full-attention output most faithfully under the fixed Top-$k$ budget. Algorithm 1 summarizes the procedure.

### 4.3. Hardware Implementation

**Hardware-Efficient Tile-sparse Kernel.** To translate the theoretical reduction in FLOPs into tangible wall-clock acceleration, we implement a highly optimized tile-skipping attention kernel leveraging the ThunderKittens DSL (Spector et al., 2024). We address the irregular memory access patterns introduced by sparse attention by exploiting the asynchronous Tensor Memory Access (TMA) and Warp

Specialization features of the NVIDIA Hopper (Markidis et al., 2018). Our kernel decouples data movement from computation via a producer-consumer paradigm. The *producer* warps orchestrate TMA instructions to fetch only the selected non-contiguous key/value tiles from global memory into a circular shared memory buffer. Simultaneously, *consumer* warps execute tensor core operations (WGMMA) on the buffered data. This design effectively hides the latency of sparse memory gathering behind dense matrix math. Consequently, our kernel reaches approximately $80\%$ of the MFU achieved by the highly optimized FlashAttention-3 (Shah et al., 2024) on 480P, 81-frame resolution sequences (sequence length $L \approx 34$K).

**Efficient Ground-Truth Heatmap Generation.** Training the sparse predictor requires a ground-truth tile-wise heatmap derived from the full-attention distribution $\mathbf{A}$. Generating exact pooled scores without storing $\mathbf{A}$ is non-trivial because softmax normalization couples all keys within each query row. We address this with a TileLang (Wang et al., 2025) kernel that computes pooled tile scores in two passes. The first pass performs tile-wise $\mathbf{Q}\mathbf{K}^\top$ in SRAM and writes per-tile unnormalized maxima together with running row statistics to global memory. The second pass finalizes the row statistics and normalizes the cached maxima to recover exact tile-wise scores. This implementation reaches $\sim 0.9 \times$ FlashAttention-3 throughput while producing the pooled supervision, and its tile-wise independence further supports *partial query processing* by supervising only a random subset of query tiles. Such sparse supervision dramatically reduces the supervision overhead and accelerates the overall training process, all without substantially compromising the final model performance.

## 5. Experiments

**Implementation Details.** We evaluate Veda using the Waver-T2V (1B/12B) (Zhang et al., 2025d) and Wan2.1-T2V (1.3B/14B) (WanTeam, 2025) architectures. To assess scalability, we conduct experiments across varying resolutions and token counts: 480P (81 frames, 34,020 tokens) for the 1B/1.3B models; 720P (81 frames, 75,600 tokens) for Wan2.1-14B; and 720P (241 frames, 219,600 tokens) for Waver-T2V-12B. All models are trained on an internal dataset encompassing a broad spectrum of general scenes. For Veda, we employ a two-stage training protocol. In the first stage, the backbone is frozen while the projector is optimized for 1,000 steps using Xavier uniform initialization and a learning rate of $6 \times 10^{-4}$ to align the sparse predictor. In the second stage, we unfreeze all parameters and perform sparse fine-tuning at the target sparsity level using a learning rate of $6 \times 10^{-5}$ for backbone and $6 \times 10^{-4}$ for sparse predictor. Unlike prior dynamic sparse methods that require a handcrafted, slow sparsity warm-up to prevent

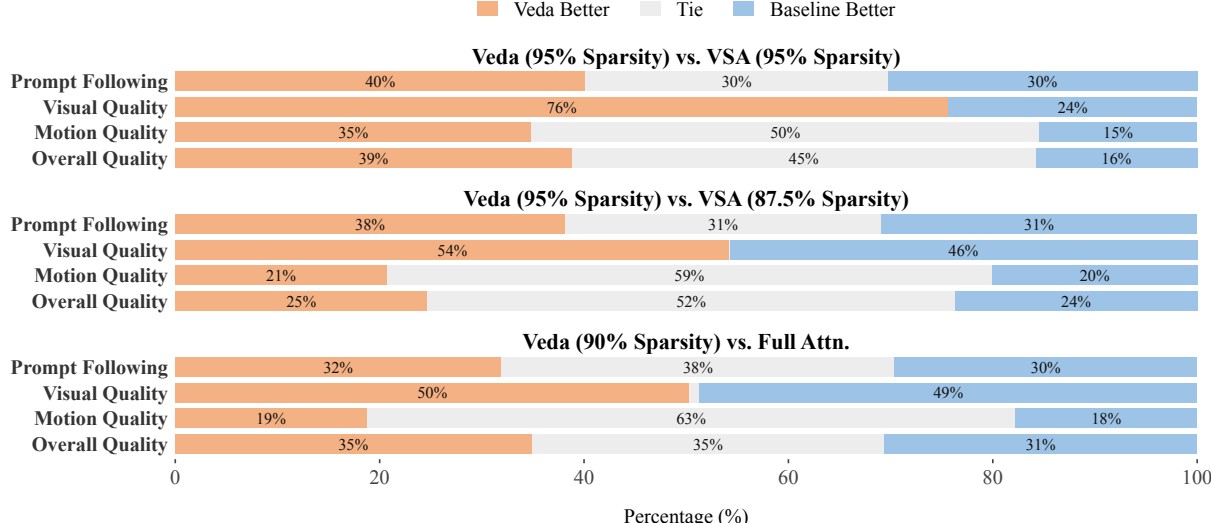

*Figure 7.* **Human evaluation on Waver-bench 1.0 using Waver-T2V-1B.** We compare **Veda** against **Full Attention** and **VSA**. Videos are generated at 480P, 81 frames (sequence length ≈ 34k). Veda achieves a parity win/tie rate against Full Attention at 90% sparsity. Notably, Veda at 95% sparsity yields superior results compared to VSA even when VSA operates at a significantly lower sparsity (87.5%), and demonstrates a substantial performance margin over VSA at equivalent (95%) sparsity levels.

training collapse, Veda's Stage 1 explicitly decouples mask learning from feature learning via a stop-gradient operation. This provides a highly stable initialization, allowing Stage 2 to rapidly adapt to target sparsity levels without fragile scheduling, and we therefore maintain the sparsity ratio at a fixed value without warmup across both stages. For experiments at the 1B/1.3B scale, models are trained for 23k steps, and we report results using Exponential Moving Average (EMA) weights with a decay of 0.9999.

**Evaluation Metrics.** We assess generation performance through a combination of human evaluation and standardized benchmarks. For human evaluation, we utilize Waver-bench 1.0 (Zhang et al., 2025d), a curated dataset of 304 prompts that cover a wide range of scenarios. Complementary to human ratings, we report quantitative metrics evaluated on the VBench (Huang et al., 2024) suite. To ensure fair comparison, all latency and speedup measurements are conducted on identical GPU hardware; dense baselines use FlashAttention-3, and sparse methods use the same tile-skipping sparse attention kernel.

**Comparison with Existing Baselines.** We benchmark Veda against Full Attention, Oracle mask and VSA (Zhang et al., 2025b). To ensure a fair comparison, all methods are trained on the identical dataset and number of steps as our approach, utilizing the specific warm-up settings proposed in their respective literature. Similarly, for all post-training sparse baselines, we strictly adhere to an equivalent training budget and learning rate schedule.

## 5.1. Generation Quality Comparison

We evaluate generation quality via a blinded Side-by-Side (SBS) study on Waver-bench 1.0 across four dimensions: *Overall Quality* (OQ), *Motion Quality* (MQ), *Visual Quality* (VQ), and *Prompt Following* (PF). Results are shown in Fig. 7.

Comparing Veda (90% sparsity) to the Full Attention base-

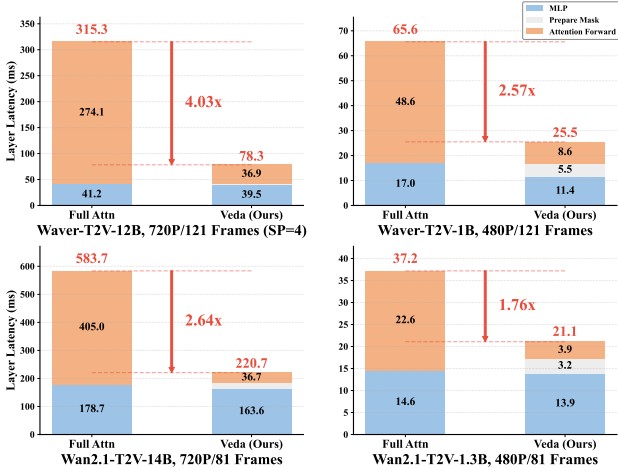

*Figure 8.* **Wall-clock latency decomposition and speedup analysis.** We measure the end-to-end per-layer inference latency of Veda against a FlashAttention-3 baseline on Waver-T2V and Wan2.1-T2V at two resolutions (720p and 480p). Stacked bars break down the runtime into MLP computation, sparse mask preparation (*Prepare Mask*), and attention kernel execution. Veda delivers substantial speedups, reaching up to 4.03× on Waver-T2V and 2.64× on Wan2.1-T2V at 720p.

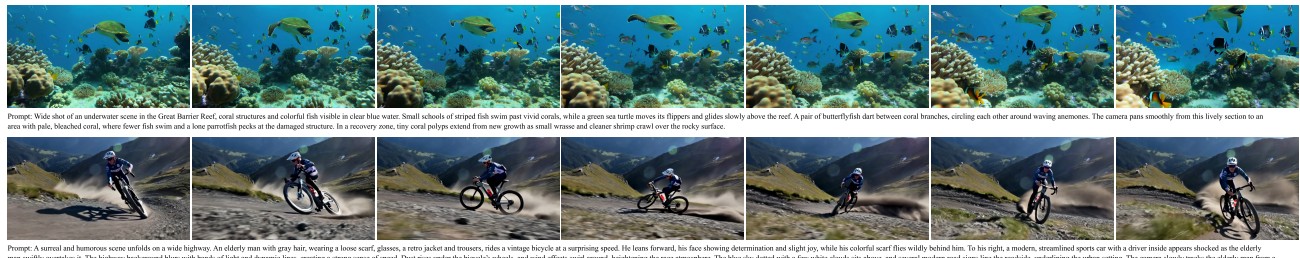

*Figure 9.* **Qualitative results of Waver-T2V-12B with 95% sparsity using Veda.** Two representative samples are shown, each synthesized at 720P resolution with 241 frames. The text prompt for each sample is shown beneath the corresponding image. More qualitative comparisons are included in the supplementary material.

*Table 1.* **VBench evaluation of Veda against full attention and state-of-the-art sparsification methods on video diffusion models.** We compare Veda against Full Attention and state-of-the-art sparsification techniques on Waver-T2V and Wan2.1-T2V. **Sparsity** indicates the reduction in computation. "Full Attn." serves as the dense baseline. ↑ and ↓ denote better performance.

| Model | Config | Method | Subject Cons. ↑ | Background Cons. ↑ | Motion Smooth. ↑ | Dynamic Deg. ↑ | Aesthetic Quality ↑ | Image Quality ↑ | Sparsity ↑ | FLOPs ↓ | Wall Time ↓ |
|---|---|---|---|---|---|---|---|---|---|---|---|
| Waver 1.0 | 1B 480P 81F | Full Attn. | 0.938 | 0.955 | 0.979 | 0.969 | 0.655 | 0.693 | 0% | $4.83 \times 10^{16}$ | 69.3s |
| | | Oracle Mask | 0.907 | 0.950 | 0.974 | 0.993 | 0.639 | 0.685 | 90% | – | 126.3s |
| | | VSA | 0.933 | 0.949 | 0.978 | 0.983 | 0.648 | 0.692 | 87.5% | $1.56 \times 10^{16}$ | 34.3s |
| | | **Ours** (S=90%) | 0.940 | 0.954 | 0.980 | 0.963 | 0.650 | 0.699 | 90% | $1.47 \times 10^{16}$ | 31.9s |
| | | **Ours** (S=95%) | 0.934 | 0.951 | 0.978 | 0.979 | 0.649 | 0.698 | 95% | $1.28 \times 10^{16}$ | 30.6s |
| Wan2.1 | 1.3B 480P 81F | Full Attn. | 0.940 | 0.969 | 0.977 | 0.844 | 0.629 | 0.670 | 0% | $5.82 \times 10^{16}$ | 58.5s |
| | | Oracle Mask | 0.957 | 0.968 | 0.931 | 0.861 | 0.325 | 0.642 | 90% | – | 94.6s |
| | | VSA | 0.911 | 0.950 | 0.975 | 0.795 | 0.549 | 0.661 | 87.5% | $2.55 \times 10^{16}$ | 42.5s |
| | | **Ours** (S=90%) | 0.887 | 0.941 | 0.972 | 0.913 | 0.543 | 0.663 | 90% | $2.46 \times 10^{16}$ | 37.6s |
| | | **Ours** (S=95%) | 0.790 | 0.926 | 0.928 | 0.733 | 0.394 | 0.661 | 95% | $2.27 \times 10^{16}$ | 35.8s |

line, we observe perceptual parity. Preference rates for VQ and PF are nearly identical (e.g., 50% vs. 49% for VQ), while a 63% tie rate in MQ confirms temporal consistency is maintained. Consequently, OQ remains indistinguishable from the full model, proving Veda preserves generative capabilities despite aggressive pruning.

Against VSA (Zhang et al., 2025b), Veda excels even under stricter constraints. Veda (95% sparsity) surpasses VSA (87.5% sparsity) in VQ (54% vs. 46%). At equal 95% sparsity, Veda dominates VQ (76% vs. 24%) and OQ (39% vs. 16%). This validates that our distillation strategy retains information-dense regions better than prior heuristic pooling methods.

In addition to human evaluation, we provide a quantitative analysis using the VBench suite in Table 1. The results demonstrate that Veda effectively retains key generative capabilities despite the significant reduction in computation. On the Waver-T2V architecture, our method maintains subject consistency and motion smoothness scores that are comparable to the Full Attention baseline, even at 95% sparsity. Across differing model scales, Veda consistently exhibits a favorable efficiency-quality trade-off relative to the VSA baseline, delivering competitive metric performance while achieving lower FLOPs and reduced wall-clock latency.

## 5.2. Inference Efficiency Comparison

We analyze wall-clock latency on NVIDIA Hopper GPUs to quantify practical speedups over FlashAttention-3 (FA3) (Shah et al., 2024). We report the end-to-end latency of a single Transformer layer, including both the sparse mask predictor overhead and the sparse attention kernel execution.

In Fig. 8, for Waver-T2V-12B at 720P with 121 frames, a dense Transformer layer takes 315.3 ms, while Veda reduces it to 78.3 ms (4.03×). The Veda breakdown is 39.5 ms for MLP computation, 1.9 ms for sparse mask preparation, and 36.9 ms for sparse attention execution, so the combined sparse-attention overhead remains well below dense attention. We observe a similar trend on Wan2.1-T2V-14B at 720P with 81 frames: layer latency decreases from 583.7 ms to 220.7 ms (2.64×), demonstrating consistent gains across backbones with different architectures and sequence configurations.

We further evaluate scalability with sequence length. As shown in Fig. 1, at 50,220 tokens (480P, 121 frames), Veda achieves a 2.57× speedup over FA3 (25.5 ms vs. 65.6 ms). When scaling to 245,760 tokens (720P, 241 frames, SP=8), FA3 grows quadratically to 1576.5 ms, whereas Veda scales near-linearly and remains at 309.1 ms, yielding

a $5.1\times$ speedup. Overall, these results confirm that Veda mitigates the quadratic attention cost and delivers robust wall-clock improvements for high-resolution, long-sequence video generation.

### 5.3. Ablation Study

**Ablation on Head-aware Tiling.** We compare various static tile configurations against our Head-aware Tiling strategy. The static baselines include configurations emphasizing spatial granularity ($[8, 8, 2]$), temporal granularity ($[4, 4, 8]$), and alternative spatial-temporal trade-offs ($[4, 8, 4]$). Among these fixed configurations, $[4, 4, 8]$ demonstrates the strongest performance and serves as the primary static baseline for comparison. We evaluate our Head-aware Tiling method against this optimal static configuration ($[4, 4, 8]$). As shown in Fig. 10, our approach achieves consistent improvements across all metrics, with notable gains of +7.2% in *Motion Quality* and +9.6% in *Overall Quality*. These results demonstrate the superiority of head-aware tiling over uniform static tiling strategies.

*Table 2.* Ablation on tile statistics for the score estimator. Results report the average training loss. *Triplet pooling* (mean, maximum, minimum) best reconstructs the oracle attention structure.

| Pooling Strategy | Training Loss ($\downarrow$) |
|---|---|
| MaxMin + Projector | 0.982 |
| Avg + Projector | 0.965 |
| **Triplet + Projector (Ours)** | **0.912** |

**Ablation on Tile Score Estimator.** We evaluate three pooling schemes for compressing token-wise features into tile-level representations: (1) *Avg*, (2) *MaxMin* (concatenating maximum and minimum), and (3) *Triplet* (concatenating mean, maximum, and minimum). We train the score estimator with the diffusion backbone frozen and report the mean squared error (MSE) to oracle full-attention tile scores. As shown in Table 2, *Triplet* achieves the lowest MSE (0.912), outperforming *Avg* and *MaxMin*.

### 5.4. Qualitative Results

Fig. 9 illustrates qualitative generation results of Waver-T2V-12B at 720p resolution for 241-frame sequences under 95% sparsity. Despite the extreme sparsity (95%), Veda maintains high visual quality and coherent motion, without noticeable sparse-induced structural artifacts (e.g., warping, water-ripple patterns, or temporal flickering). More qualitative examples are included in the Supplementary Material.

## 6. Conclusion and Future Work

We presented Veda, a hardware-aware sparse attention framework that addresses the scalability bottleneck of high-resolution video Diffusion Transformers. Veda reformulates

sparsity as an explicit tile selection problem and supervises tile scoring via full-attention distillation, rather than relying on the diffusion objective to implicitly shape sparse patterns. The induced Top-$k$ masks preserve the tile-wise structure and ranking of full attention even under extreme pruning. By combining a statistics-aware tile score estimator and Head-Aware Tiling with a custom tile-skipping kernel, Veda translates algorithmic sparsity into practical wall-clock gains, achieving up to $5.1\times$ end-to-end speedup on Waver and Wan2.1 with no noticeable degradation in generation quality, while exhibiting favorable scaling as spatiotemporal sequence length increases. Looking forward, further improvements may come from tighter kernel-level fusion to reduce mask preparation overhead, more advanced scheduling to sustain high MFU at sparsity levels beyond 95%, and richer distillation signals or adaptive sparsity strategies that allocate computation across timesteps and heads. Incorporating lightweight temporal caching of tile scores across adjacent diffusion steps could further improve stability.

## Impact Statement

This paper presents work whose goal is to advance the field of Machine Learning. There are many potential societal consequences of our work, none of which we feel must be specifically highlighted here.

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

# A. Theoretical Framework for Veda

In this section, we establish the theoretical foundations of our approach and provide deeper algorithmic and mathematical insights into the nature of sparse attention.

## A.1. The Necessity of Accurate Oracle Masks

The self-attention mechanism can be formally viewed as an Energy-Based Model (EBM)(Ramsauer et al., 2021; Litman, 2025). Let $\mathbf{u} \in \mathbb{R}^N$ denote the logits (pre-softmax scores), where $u_j = \mathbf{q}^T \mathbf{k}_j$. The generating potential function of the attention distribution is the Log-Sum-Exp (LSE) function(Litman, 2025), $\Psi(\mathbf{u})$:

$$\Psi(\mathbf{u}) = \tau \log \sum_{j=1}^{N} \exp(u_j/\tau), \tag{10}$$

where $\tau$ is the temperature parameter. The attention probability distribution $P$ is the gradient of this potential, $P = \nabla \Psi(\mathbf{u})$. The LSE function is dominated by the maximum term, $u_{max}$.

**Proposition A.1** (Distributional Shift via Mask Error). *Let $\mathcal{M}^*$ be the set of indices corresponding to the true Top-k keys (Oracle), and let $\hat{\mathcal{M}}$ be an estimated mask such that $\arg\max(\mathbf{u}) \notin \hat{\mathcal{M}}$. The perturbed potential $\tilde{\Psi}$ excludes the dominant energy term. This exclusion forces a renormalization of probability mass over the retained keys $\hat{\mathcal{M}}$:*

$$\tilde{P}_j = \frac{\exp(u_j/\tau)}{\sum_{r \in \hat{\mathcal{M}}} \exp(u_r/\tau)} = \frac{Z}{\hat{Z}} P_j > P_j, \quad \forall j \in \hat{\mathcal{M}}, \tag{11}$$

*where $Z = \sum_{r=1}^{N} \exp(u_r/\tau)$ and $\hat{Z} = \sum_{r \in \hat{\mathcal{M}}} \exp(u_r/\tau)$.*

This renormalization can assign inflated confidence to noise or irrelevant tokens, a phenomenon we term *structural hallucination*. Unlike linear operations, where errors are additive, the softmax nonlinearity can amplify mask errors through exponentiated logits and row-wise renormalization. Thus, accurate recovery of the Oracle Mask is a structural prerequisite for high-sparsity attention.

## A.2. Limitations of Linear Pooling Representations

Existing methods like VSA(Zhang et al., 2025b) or Sparge Attention(Zhang et al., 2025a) often employ Average Pooling to compress tiles. We argue that linear pooling is theoretically insufficient for approximating the bilinear attention operation. From a spectral perspective, Average Pooling is a low-pass filter equivalent to a projection matrix $W_{avg}$. The approximated score becomes $\hat{u}_{ij} = \mathbf{q}_i^T W_{avg}^T W_{avg} \mathbf{k}_j$.

**Lemma A.2** (Loss of Second-Order Statistics). *The inner product $\langle \mathbf{q}, \mathbf{k} \rangle$ depends on both the angle and magnitude of the high-frequency components of the vectors. The projection $W_{avg}$ collapses the vector variance onto the mean, discarding the high-frequency harmonics required to distinguish orthogonal vectors. Mathematically, $\langle \bar{\mathbf{q}}, \bar{\mathbf{k}} \rangle \neq \overline{\langle \mathbf{q}, \mathbf{k} \rangle}$. The loss of orthogonality leads to rank deficiency in the approximated attention matrix, making it impossible to distinguish between distinct but mean-similar tiles.*

## A.3. Reconstructing Attention via Sufficient Statistics

To reconstruct the attention distribution accurately with minimal I/O, we propose a statistical estimator based on vector decomposition. We treat the elements of dimension-$d$ vectors $\mathbf{q}$ and $\mathbf{k}$ as samples from a distribution.

**Theorem A.3** (Dot Product Decomposition). *The dot product of two vectors $\mathbf{q}, \mathbf{k} \in \mathbb{R}^d$ can be decomposed into a mean product term and a covariance term:*

$$\mathbf{q} \cdot \mathbf{k} = d\mu_q\mu_k + d\operatorname{Cov}(\mathbf{q}, \mathbf{k}), \tag{12}$$

*where $\operatorname{Cov}(\mathbf{q}, \mathbf{k}) = \rho\sigma_q\sigma_k$.*

While $\mu$ (Avg) captures the centroid, reconstructing the covariance requires information about the spread ($\sigma$) and correlation ($\rho$). We employ the Bhatia-Davis inequality to bound the variance using only extremum statistics.

**Lemma A.4** (Variance Bound via Extrema). *For any bounded random variable $X$ with support $[m, M]$ and mean $\mu$, the variance is bounded by:*

$$\sigma^2 \leq (M - \mu)(\mu - m). \tag{13}$$

Assuming the vector elements follow a unimodal distribution (e.g., Gaussian or Laplacian, typical in LayerNorm-normalized embeddings), the standard deviation is proportional to the range: $\sigma \approx \frac{M-m}{\alpha}$. By retaining the triplet $\{\mu, m, M\}$ (Avg, Min, Max), we effectively construct a *hyper-rectangle bounding box* that constrains the manifold of the vectors. We derive our estimator $\mathcal{S}(\mathbf{q}, \mathbf{k})$ as:

$$\mathcal{S} \approx d \cdot \mu_q \mu_k + \lambda \Delta_q \Delta_k \tag{14}$$

where $\Delta_x = \sqrt{(M_x - \mu_x)(\mu_x - m_x)}$ represents the dispersion bound for $x \in \{q, k\}$. Here, the first term anchors the centroid interaction, while the second term approximates the maximal energy potential of the covariance.

### A.4. The Necessity of Intra-Tile Similarity

We now justify why tokens within a tile should be similar for this approximation to hold. Consider tile-sparse attention where a single statistic represents a tile of size $B$. Let $\boldsymbol{\mu_q}$ and $\boldsymbol{\mu_k}$ denote the tile means, and define the residual vectors $\mathbf{r}_i^q = \mathbf{q}_i - \boldsymbol{\mu_q}$ and $\mathbf{r}_i^k = \mathbf{k}_i - \boldsymbol{\mu_k}$. The approximation error $E$ from the neglected residual interactions can be written as:

$$E = \left| \sum_{i=1}^{B} \langle \mathbf{r}_i^q, \mathbf{r}_i^k \rangle \right|. \tag{15}$$

By the Cauchy-Schwarz inequality, this error is bounded by the product of the tile-wise standard deviations:

$$E \leq \left( \sum_{i=1}^{B} \|\mathbf{r}_i^q\|_2^2 \right)^{1/2} \left( \sum_{i=1}^{B} \|\mathbf{r}_i^k\|_2^2 \right)^{1/2} = B \cdot \sigma_{\mathbf{q}} \cdot \sigma_{\mathbf{k}}, \tag{16}$$

where $\sigma_{\mathbf{q}}^2 = \frac{1}{B} \sum_{i=1}^{B} \|\mathbf{r}_i^q\|_2^2$ and $\sigma_{\mathbf{k}}^2 = \frac{1}{B} \sum_{i=1}^{B} \|\mathbf{r}_i^k\|_2^2$.

**Corollary A.5** (Clustering Condition). *A sufficient condition for the sparse approximation to be robust (i.e., $E \to 0$) is $\sigma_{\mathbf{q}} \to 0$ or $\sigma_{\mathbf{k}} \to 0$, and reducing both variances tightens the bound. This implies that tokens within a tile should lie on a local low-dimensional manifold with minimal variance. If tiles contain diverse tokens (high $\sigma$), the uncertainty bound of the estimator grows, rendering the sparse mask unreliable.*

### A.5. Token Clustering as Bandwidth Minimization

To satisfy the condition $\sigma \to 0$, we formulate token clustering as a graph labeling problem. Let $\mathcal{G} = (\mathcal{V}, \mathcal{E})$ be the token graph where edge weights $W_{ij} \propto \exp(\mathbf{q}_i^T \mathbf{k}_j)$. We seek a bijective mapping $\phi : \mathcal{V} \to \{1, \ldots, N\}$ that minimizes the matrix bandwidth.

**Definition A.6** (Manifold Embedding under 3D RoPE). Let tokens $x_i$ reside on a 3D spatio-temporal manifold $\mathcal{M}$ induced by Rotary Positional Embeddings (RoPE). The attention affinity decays with the geodesic distance on this manifold: $W_{ij} \approx f(\|\mathbf{x}_i - \mathbf{x}_j\|_\Sigma)$. A locality-preserving mapping $\phi$ must satisfy the Hölder continuity condition:

$$|\phi(u) - \phi(v)| \leq C \cdot \|\mathbf{x}_u - \mathbf{x}_v\|_2^\alpha. \tag{17}$$

Standard raster scanning ($\phi_{\text{flat}}$) violates this condition for temporal neighbors, yielding a Lipschitz constant $C \propto H \times W$, resulting in a dispersed, high-bandwidth attention matrix. We propose **Head-aware 3D Tiling**, which creates a mapping $\phi_{\text{tile}}$ where temporal neighbors satisfy $|\phi_{\text{tile}}(u) - \phi_{\text{tile}}(v)| \leq p_h p_w$. This effectively folds the 3D manifold into 1D memory while preserving local neighborhoods, implicitly performing graph clustering. Furthermore, we employ *Anisotropic Metric Adaptation* to search for optimal tile shapes $(p_t, p_h, p_w)$, aligning the tiling strategy with the specific induced metric $d_h(\mathbf{x}_i, \mathbf{x}_j)$ of each attention head. This topological optimization minimizes attention bandwidth, effectively clustering high-affinity tokens to lower intra-tile variance $\sigma$. Consequently, the geometric regularization satisfies the spectral approximation conditions, validating the representational power of the compressed statistics $\{\mu, m, M\}$.

### A.6. Projector Optimization via Information Geometry

Finally, we address the non-differentiability of the Top-$k$ selection required to train the MLP projector. We leverage the framework of Entropic Optimal Transport (EOT) (Litman, 2025). The attention mechanism operates on a statistical manifold equipped with the Fisher-Rao metric. The standard Euclidean gradient is suboptimal for optimization on the probability simplex.

**Theorem A.7** (Local KL Geometry). *Let $P_{\text{oracle}}$ be the ground truth distribution derived from full attention, and $P_{\text{pred}}$ be the distribution predicted by the MLP. Around the oracle parameters $\theta_t$, the Kullback-Leibler (KL) divergence admits the local second-order approximation:*

$$\mathcal{L} = D_{\text{KL}}(P_{\text{oracle}} \| P_{\text{pred}}) \approx \frac{1}{2}(\theta_t - \theta_s)^T I(\theta_t)(\theta_t - \theta_s). \tag{18}$$

*Thus, KL supervision locally weights prediction errors by the Fisher Information Matrix $I(\theta_t)$, matching the natural geometry of the probability simplex.*

By utilizing KL divergence as the training objective, we train the score estimator before the discrete Top-$k$ operation and therefore avoid differentiating through Top-$k$. Instead, we force the MLP to learn the *geometry of the energy landscape* (the LSE potential field). This encourages the predicted logits $\hat{\mathbf{u}}$ to maintain the same ranking and local geometry as the oracle logits $\mathbf{u}$, leading to accurate mask recall during inference.

## B. Detailed Implementation Algorithms for Veda

We provide the complete algorithmic pseudocode for the core components of Veda, which are omitted from the main text due to space constraints below.

---

**Algorithm 1** Head-Aware Tiling Search

---

1: **Input:** Latent dims $(T, H, W)$, Calibration Set $\mathcal{D}_{\text{cal}}$, Tile Size $B = 128$, Top-$k$ Budget $k_{\text{top}}$.
2: **Output:** Optimal Configuration Map $\boldsymbol{\pi}^* \in \Omega^{L \times N_h}$.
3:   // Construct hardware-aligned search space
4: $\Omega \leftarrow \{(p_t, p_h, p_w) \in \mathbb{N}^3 \mid p_t p_h p_w = B\}$
5: Initialize cumulative error $\mathcal{E} \in \mathbb{R}^{L \times N_h \times |\Omega|}$ to $\mathbf{0}$.
6: **for** sample $x \in \mathcal{D}_{\text{cal}}$ **do**
7:     Perform inference to cache $\{\mathbf{Q}, \mathbf{K}, \mathbf{V}\}$
8:     **for** layer $l \in \{1 \ldots L\}$, head $h \in \{1 \ldots N_h\}$ **do**
9:       // Get Full-Attention Map and Output
10:       $\mathbf{A}^{\text{fu}}_{l,h} \leftarrow \text{Softmax}(\mathbf{Q}_{l,h}\mathbf{K}^{\top}_{l,h}/\sqrt{d})$
11:       $\mathbf{O}^{\text{fu}}_{l,h} \leftarrow \mathbf{A}^{\text{fu}}_{l,h}\mathbf{V}_{l,h}$
12:       **for** $\pi \in \Omega$ **do**
13:         // 1. Tiling QKV and Full-Attention Map
14:         $\tilde{\mathbf{Q}}, \tilde{\mathbf{K}}, \tilde{\mathbf{V}} \leftarrow \text{Tiling}(\{\mathbf{Q}_{l,h}, \mathbf{K}_{l,h}, \mathbf{V}_{l,h}\}, \pi)$
15:         $\mathbf{A}^{\pi} \leftarrow \text{Softmax}(\tilde{\mathbf{Q}}\tilde{\mathbf{K}}^{\top}/\sqrt{d})$
16:         // 2. Generate Oracle Mask from Full Attention
17:         $\mathbf{S}^{\text{tile}} \leftarrow \text{RowNorm}(\text{MaxPool}(\mathbf{A}^{\pi}))$
18:         $\widetilde{\mathbf{M}} \leftarrow \text{Top-}k(\mathbf{S}^{\text{tile}}, k_{\text{top}})$
19:         // 3. Compute Sparse Output and Error
20:         $\mathbf{O}^{\text{sp}}_{l,h} \leftarrow \text{UnTile}(\text{SparseAttn}(\tilde{\mathbf{Q}}, \tilde{\mathbf{K}}, \tilde{\mathbf{V}}, \widetilde{\mathbf{M}}), \pi)$
21:         $\mathcal{E}_{l,h,\pi} \leftarrow \mathcal{E}_{l,h,\pi} + \|\mathbf{O}^{\text{fu}}_{l,h} - \mathbf{O}^{\text{sp}}_{l,h}\|^2_F$
22:       **end for**
23:     **end for**
24: **end for**
25: // Select configuration minimizing expected error
26: $\pi^*_{l,h} \leftarrow \arg\min_{\pi \in \Omega} \mathcal{E}_{l,h,\pi} \quad \forall l, h$
27: **return** $\boldsymbol{\pi}^*$

---

---

**Algorithm 2** Tile Score Estimation & Mask Generation

---

1: **Input:** Query $\mathbf{Q}$, Key $\mathbf{K} \in \mathbb{R}^{N_h \times N \times d}$, Tile Size $B$, Top-$k$ Budget $k_{\text{top}}$, Latent Dim $d'$
2: **Output:** Predicted distribution $\mathbf{A}^{\text{pred}}$ and binary mask $\widetilde{\mathbf{M}} \in \{0, 1\}^{N_h \times T_q \times T_k}$
3: // Tiling
4: $T_q \leftarrow N/B, \quad T_k \leftarrow N/B$
5: $\tilde{\mathbf{Q}} \leftarrow \text{Tile}(\mathbf{Q}, \{N_h, T_q, B, d\})$
6: $\tilde{\mathbf{K}} \leftarrow \text{Tile}(\mathbf{K}, \{N_h, T_k, B, d\})$
7: // Triplet Statistics Extraction (TripPool)
8: $\mathbf{Z}_q \leftarrow \text{Avg}[\tilde{\mathbf{Q}}] \oplus \text{Max}[\tilde{\mathbf{Q}}] \oplus \text{Min}[\tilde{\mathbf{Q}}]$
9: $\mathbf{Z}_k \leftarrow \text{Avg}[\tilde{\mathbf{K}}] \oplus \text{Max}[\tilde{\mathbf{K}}] \oplus \text{Min}[\tilde{\mathbf{K}}]$
10: // Projection & Score Estimation
11: **for** each head $h \in \{1, \ldots, N_h\}$ **do**
12: $\quad \hat{\mathbf{q}}_h \leftarrow \phi_q^{(h)}(\mathbf{Z}_{q,h}), \quad \hat{\mathbf{k}}_h \leftarrow \phi_k^{(h)}(\mathbf{Z}_{k,h})$
13: $\quad \mathbf{S}_h^{\text{pred}} \leftarrow \frac{\hat{\mathbf{q}}_h \hat{\mathbf{k}}_h^\top}{\sqrt{d'}}$
14: $\quad \mathbf{A}_h^{\text{pred}} \leftarrow \text{Softmax}(\mathbf{S}_h^{\text{pred}})$             // row-wise over key tiles
15: $\quad \widetilde{\mathbf{M}}_h \leftarrow \mathbb{I}(\mathbf{S}_h^{\text{pred}} \geq \text{Top-}k\text{-Threshold}(\mathbf{S}_h^{\text{pred}}, k_{\text{top}}))$
16: **end for**
17: **return** $\mathbf{A}^{\text{pred}}, \widetilde{\mathbf{M}}$

---

**Algorithm 3** Tile-Wise Distillation for Tile Score Estimator

---

1: **Input:** Query $\mathbf{Q}$, Key $\mathbf{K} \in \mathbb{R}^{N_h \times N \times d}$, Tile Size $B$, Top-$k$ Budget $k_{\text{top}}$, Latent Dim $d'$
2: **Output:** Distillation Loss $\mathcal{L}_{\text{distill}}$
3: // Full-Attention Teacher
4: $\mathbf{A}^* \leftarrow \text{Softmax}\left(\frac{\mathbf{Q}\mathbf{K}^\top}{\sqrt{d}}\right)$
5: // Target Construction (Score & Weight)
6: $\mathbf{S}^{\text{tgt}} \leftarrow \text{MaxPool}(\mathbf{A}^*, \text{kernel} = B, \text{stride} = B)$
7: $\mathbf{A}^{\text{tgt}} \leftarrow \text{RowNormalize}(\mathbf{S}^{\text{tgt}})$
8: // Student Prediction (Estimator with Stop-Gradient)
9: $\mathbf{A}^{\text{pred}}, \widetilde{\mathbf{M}} \leftarrow \text{Algorithm } 2(\text{sg}(\mathbf{Q}), \text{sg}(\mathbf{K}), B, k_{\text{top}}, d')$
10: // Distribution Alignment
11: $\mathcal{L}_{\text{distill}} \leftarrow \text{Mean}_{h,i}\left[\mathcal{D}_{\text{KL}}\left(\mathbf{A}^{\text{tgt}}[h, i, :] \parallel \mathbf{A}^{\text{pred}}[h, i, :]\right)\right]$
12: **backward** $\mathcal{L}_{\text{distill}}$

---

## C. Waver-T2V Implementation Details

This section provides specifications for both the Waver-T2V-1B and Waver-T2V-12B models (Zhang et al., 2025d). The tables below detail the architectural configurations and training hyperparameters. The 1B variant employs a Multimodal Diffusion Transformer (MM-DiT) (Yang et al., 2025) backbone with 8 Dual Flow blocks and 22 Single Flow blocks. The 12B variant scales to 16 Dual Flow layers and 40 Single Flow layers with expanded dimensionality. Both models utilize the Wan-VAE for latent representation and Flan-T5-XXL for text encoding.

*Table 3.* Model configuration comparison between Waver-T2V-1B and Waver-T2V-12B. Both architectures utilize a hybrid backbone comprising Dual Flow and Single Flow Multimodal Diffusion Transformer (MM-DiT) blocks.

| Waver-T2V-1B | | Waver-T2V-12B | |
|---|---|---|---|
| **Model Config** | **Value** | **Model Config** | **Value** |
| Model Size | 1B | Model Size | 12B |
| Backbone Architecture | MM-DiT | Backbone Architecture | MM-DiT |
| Dual Flow Layers | 8 | Dual Flow Layers | 16 |
| Single Flow Layers | 22 | Single Flow Layers | 40 |
| Patch Size | $1 \times 2 \times 2$ | Patch Size | $1 \times 2 \times 2$ |
| Hidden Dimension (Attention Head) | 128 | Hidden Dimension (Attention Head) | 128 |
| Num Heads | 12 | Num Heads | 24 |
| In Channels | 16 | In Channels | 36 |
| Out Channels | 16 | Out Channels | 16 |
| Cross Attention Dim | 1536 | Cross Attention Dim | 3072 |
| Text Embedding Dim | 4096 | Text Embedding Dim | 4096 |
| VAE Model | Wan-VAE | VAE Model | Wan-VAE |
| Activation Function | GELU | Activation Function | GELU |
| Normalization | RMS Norm | Normalization | AdaLayerNormZero |
| Positional Embedding | RoPE | Positional Embedding | RoPE |

*Table 4.* Training hyperparameters comparison between Waver-T2V-1B and Waver-T2V-12B. Both models employ Rectified Flow matching objectives with distinct optimization configurations.

| Waver-T2V-1B | | Waver-T2V-12B | |
|---|---|---|---|
| **Hyperparameter** | **Value** | **Hyperparameter** | **Value** |
| Learning Rate | $6.0 \times 10^{-5}$ | Learning Rate | $5.0 \times 10^{-6}$ |
| LR Scheduler | Constant | LR Scheduler | Constant |
| Batch Size | 128 | Batch Size | 64 |
| Resolution | $480 \times 864$ (161 frames) | Resolution | $720 \times 1280$ (241 frames) |
| Optimizer | AdamW | Optimizer | AdamW |
| AdamW $\beta$ | $(0.9, 0.999)$ | AdamW $\beta$ | $(0.9, 0.999)$ |
| Weight Decay | $1.0 \times 10^{-2}$ | Weight Decay | $1.0 \times 10^{-2}$ |
| Gradient Clipping | 1.0 | Gradient Clipping | 1.0 |
| Precision | BF16 | Precision | BF16 |
| Training Objective | Flow Matching | Training Objective | Flow Matching |

# D. Human Evaluation of Dynamic versus Static Tile Size Configurations

To empirically validate the effectiveness of our Head-aware dynamic tile size selection strategy, we conduct a human evaluation study comparing various static tile configurations against our proposed approach, where human annotators assess the perceptual quality and generation fidelity of the outputs.

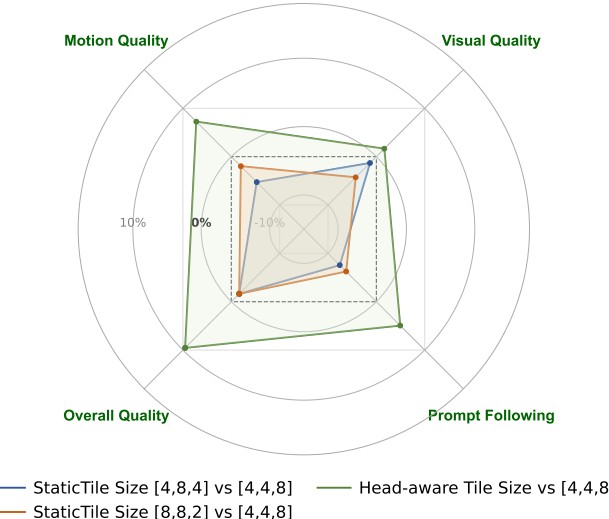

*Figure 10.* **Human evaluation of dynamic versus static tile size configurations.** We benchmark various static tile size settings alongside our proposed Head-aware dynamic selection method against the strongest static baseline ($[4, 4, 8]$). While the $[4, 4, 8]$ configuration emerges as the optimal fixed setting, our dynamic approach consistently outperforms all static alternatives, achieving positive net win rates across evaluations and demonstrating superior generation quality through adaptive tile size allocation.

