# OpenReview forum: "Veda: Scalable Video Diffusion via Distilled Sparse Attention"
_ICML.cc/2026/Conference — ICML 2026 regular_

### Official Review · Reviewer_Ekm9 · 2026-03-08

**Soundness:** 3
**Presentation:** 3
**Significance:** 3
**Originality:** 3
**Overall Recommendation:** 4
**Confidence:** 3

**Summary:**

The authors propose a distilled sparse attention framework Veda, which formulates tile selection as an explicit reconstruction problem from full attention. By explicitly reconstructing tilewise attention masks using statistics-aware scoring and head-aware tiling, Veda achieves up to a 5.1x wall-clock e2e speedup without degrading visual quality at up to 95% sparsity.

**Compliance With Llm Reviewing Policy:**

Affirmed.

**Final Justification:**

The authors addressed the concerns on hardware requirements and model generalization, the authors also answers the questions on training overheads and wordings. I increased my rating to 4.

**Key Questions For Authors:**

Please refer to weakness.

**Limitations:**

Yes

**Strengths And Weaknesses:**

Strengths:

- Veda effectively tackles the quadratic computation and memory complexity of self-attention.
- Instead of relying on the standard diffusion objective to implicitly learn sparse structures, Veda frames tile selection as an explicit objective and preserves the tile-wise structure of full attention.
- The paper is well structured and the ablation studies are thorough.

Weakness:

- The 5.1x wall-clock speedup is hardware-dependent. Section 4.3 mentions that the custom kernel relies on NVIDIA Hopper-specific features. Testing on different specifications would help generalizing the observation.
- Lack of thorough comparisons in table2: only compared with VSA.
- The performance in Table 2 is not consistent for Wan and Waver. for example, Waver gets better score on subject const. at 90% sparsity while Wan becomes worse than VSA. Would be better to test more models for more consistent performance.
- L194 mentioned Veda as "plug-and-play module", but it still requires a computationally expensive two-stage training protocol.
- Veda uses a lightweight estimator to predict where to attend, mask prediction step takes a fixed chunk of time (e.g., ~39 ms per layer on a 12B model). If running at a lower sparsity (e.g., at a safer rate like 30%), the overhead would likely make it slower than just running standard Full Attention.

---

> ### Author Rebuttal · Authors · 2026-03-31
>
> We sincerely thank the reviewer for the constructive feedback. We address each concern below.
>
> ---
>
> **Q1: Hardware Generalizability of Speedup**
>
> We appreciate this important point. We clarify that Veda's tile-skipping mechanism is **architecturally orthogonal** to FlashAttention — the Hopper-specific features (TMA, warp specialization) are implementation optimizations for hiding irregular memory access latency, not algorithmic requirements. Veda's sparse mask prediction (Sec. 4.1) and tile selection (Sec. 4.2) are entirely **hardware-agnostic**. The tile-skipping kernel only requires selectively loading KV tiles conditioned on a binary mask — a capability supported by any GPU with shared memory and block-level control flow. On Ampere (A100), one would replace TMA with standard `cp.async` global-to-shared copies and warp specialization with a conventional pipelined loop, following the FlashAttention-2 paradigm.
>
> **Architecture-independent scaling.** Since Veda's speedup derives from reducing processed tiles from $N_T^2$ to $\sim K \cdot N_T$, the theoretical FLOPs reduction is hardware-independent. The wall-clock translation efficiency (our ~0.8× MFU relative to dense FA) may vary across GPU generations, but the fundamental scaling advantage is preserved. We will add Ampere benchmarks and a portability discussion in the revised manuscript.
>
> ---
>
> **Q2 & Q3: Comparison Scope and Cross-Model Consistency**
>
> **On comparison scope.** We acknowledge that Table 2 focuses primarily on VSA. Our rationale is that VSA represents the current state-of-the-art dynamic sparse attention method for video DiTs, making it the most direct comparison alongside full attention. We agree that including static-pattern approaches would strengthen the empirical picture and will add comparisons with representative methods (e.g., SVG) in the camera-ready version.
>
> **On cross-model VBench consistency.** We respectfully argue that the per-metric fluctuations the reviewer observes should not be interpreted as indicating inconsistent quality. As detailed in our response to Reviewer SQWi (W1), VBench has well-documented limited discriminative power at this quality regime — recent top-venue works consistently demonstrate that VBench lacks sufficient capacity to distinguish strong modern models, making fine-grained score differences unreliable.
>
> Our **human evaluation reveals a consistent story across both models**. On Waver-T2V-1B (95% sparsity), evaluators prefer Veda over VSA by 76% vs. 24% on Visual Quality and 39% vs. 16% on Overall Quality (Figure 7). We further conducted a supplementary SBS evaluation on Wan2.1-1.3B (64-prompt diverse subset, 90% sparsity): Veda is preferred 35.9% vs. 17.2% for VSA (46.9% ties), confirming the advantage generalizes across architectures beyond what VBench captures.
>
> We will expand model coverage in the camera-ready version where computationally feasible.
>
> ---
>
> **Q4: "Plug-and-Play" Claim and Training Cost**
>
> We acknowledge that "plug-and-play" may be misleading and will revise the phrasing. Our intended meaning is **architectural modularity**: the Veda attention module replaces dense self-attention in any DiT block without modifying surrounding components (MLP, normalization, cross-attention).
>
> Regarding training cost, Veda's two-stage protocol is designed to **minimize total convergence overhead**. In Stage 1, the entire backbone is frozen and only the lightweight score projector is optimized for 1,000 steps — negligible relative to any training budget. This rapidly aligns tile scoring to full-attention geometry. Stage 2 unfreezes the backbone for sparse fine-tuning from a well-initialized scoring function that already produces high-recall masks, substantially reducing required adaptation steps. In contrast, VSA requires a handcrafted slow warmup — gradually increasing sparsity to prevent training collapse. Veda's explicit distillation objective (Eq. 7) decouples mask learning from feature learning via stop-gradient, stabilizing training from the outset and removing such fragile scheduling. We will clarify this in the revision.
>
> ---
>
> **Q5: Overhead at Low Sparsity**
>
> The reviewer is correct that at low sparsity (e.g., 30%), the fixed mask prediction overhead would negate compute savings. We clarify that **Veda is not designed for low sparsity**. As shown in our human evaluation (Fig. 7), Veda at 90% sparsity already matches full attention in generation quality, and at 95% outperforms VSA at 87.5%. Operating at 30% would yield no quality gain while sacrificing nearly all acceleration benefit.
>
> This aligns with the broader sparse attention research goal: the key difficulty has always been **maintaining quality at high sparsity**, not achieving marginal speedups at conservative rates. Veda directly addresses this challenge, and its mask overhead is well-amortized in the high-sparsity regime — particularly as sequence length grows ($2.57\times \to 5.1\times$ speedup, Fig. 1).

---

> > ### Author Rebuttal · Reviewer_Ekm9 · 2026-04-03
> >
> > Thanks for the authors to provide more details, my concerns are fully addressed

---

> > > ### Author Response · Authors · 2026-04-07
> > >
> > > Dear Reviewer Ekm9,
> > >
> > > Thank you very much for reading our rebuttal and for your thoughtful follow-up. We truly appreciate your time and are very glad to know that our response has addressed your concerns.
> > >
> > > If you do not have further remaining concerns, we would be very grateful if you could kindly consider updating your initial rating based on the clarifications provided.
> > >
> > > If there are still any additional questions or points that would be helpful for us to clarify, we would be more than happy to address them. As the formal author response rounds are now very limited, we would still greatly appreciate the opportunity to provide clarification through the AC discussion/comments if needed.
> > >
> > > Thank you again for your time and consideration.

---

### Official Review · Reviewer_dUXi · 2026-03-08

**Soundness:** 3
**Presentation:** 3
**Significance:** 3
**Originality:** 4
**Overall Recommendation:** 4
**Confidence:** 3

**Summary:**

Quadratic cost of self-attention is the main bottleneck for scaling video generation models, this paper aims to propose a novel sparse attention.

Empirically, the authors find that mask quality, i.e., the degree to which the sparse mask aligns with the tile-wise structure of full attention, dominates performance.

Based on above observation, the proposed method employs a lightweight estimator to distill tile-level attention scores from a full-attention backbone, thus the sparisity mask can be used wisely.

A tile-skipping sparse attention kernel is implemented for real acceleration.

On Waver-T2V-12B generating 720P 10-second videos, the proposed method achieves a 5.1× end-to-end speedup and a 10.5× self-attention speedup, reducing attention overhead from 90% to 50%. On Wan2.1-T2V-14B generating 720P 5-second videos, the proposed method further delivers a 2.63× end-to-end speedup with a 7.08× acceleration in self-attention.

**Compliance With Llm Reviewing Policy:**

Affirmed.

**Final Justification:**

Most of my concerns are solved.

**Key Questions For Authors:**

(1) Will the code and the sparsity kernel be open sourced? Otherwise, it will be very difficult to reproduce the wall-clock latency improvement results.

(2) For human evaluation, the concrete procedure, the amount of people, screen settings etc. are necessary information.

(3) It is better to show the quantitative comparison results with previous works on higher resolution generation.

(4) The results in table 2 is not analyzed. It seems that the proposed method does not outperform VSA regarding those metrics on VBench.




I am willing to raise the score if my concerns can be addressed.

**Limitations:**

No.

One possible limitation is that the proposed method does not provide better video quality than VSA in Table 2.

**Strengths And Weaknesses:**

Strength:
The presentation is good and the manuscript is easy to follow.

The observation of tile-wise mask quality dominating sparsity ration is interesting, reasonable and novel to me, which is a clear motivation for the development of a tile-wise reconstruction-style masking method in this paper.

Aside from a good motivation, this paper also has good implementations in the method design, which integrates statistics-aware tile scoring with head-aware tiling to reduce estimation error and structural mismatch, enabling aggressive sparsity.

Human evaluation on Waver-bench 1.0 using Waver-T2V-1B shows that the proposed sparse attention achieves comparable video quality with full attention.



Weakness:
The details on human evaluation is not provided. The concrete procedure, the amount of people, screen settings etc. are necessary information, which should be included in the appendix.

In table 2, only 480p video generation is compared, it is not clear whether the proposed method still perform well on higher resolutions. In VSA, 720p video generation is evaluated.

---

> ### Author Rebuttal · Authors · 2026-03-31
>
> We sincerely thank Reviewer dUXi for the constructive feedback and the recognition of our tile-wise observation, method design, and human evaluation results. We address each concern below.
>
> ---
>
> **Q1: Will the code and sparsity kernel be open-sourced?**
>
> We are happy to release the full package to the community, conditioned on that the release does not violate the policies of our supporting organization. We are currently coordinating with our institution on the release scope and timeline.
> Besides the code release plan, we will provide detailed pseudocode in the supplementary material to help the community better understand and adopt our method. We believe this level of detail will enable practitioners to migrate the core idea to their own codebases.
>
> ---
>
> **Q2: Human evaluation details**
>
> We agree that the details of human evaluation are important for reproducibility and will include a comprehensive description in the appendix. We clarify the key aspects below.
>
> **Evaluation Personnel & Quality Control.** Our Side-by-Side (SBS) evaluations are conducted by the quality control (QC) department of the supporting organization. All evaluators are **highly trained employees** with professional experience in video quality assessment. Each sample is **independently scored by at least three evaluators**. The QC department will compute pairwise inter-annotator agreement (recall) among them; if the agreement falls below a pre-defined threshold, additional evaluators are recruited until a reliable consensus is reached.
>
> **Blinding & Interface Design.** The evaluation follows a **strict double-blind protocol**: (1) evaluators have no knowledge of which model produced which video; (2) both the presentation order and the left/right spatial placement of compared videos are randomized for each trial. Evaluators are provided with an interactive interface that allows them to **play, pause, speed up, and slow down** videos for fine-grained comparison. We will include a screenshot of the equivalent evaluation interface in the revised appendix.
>
> **Metrics.** We define 12 fine-grained dimensions organized into three categories — **Motion Quality** (action naturalness, interaction plausibility, temporal distortion), **Visual Quality** (image quality, color, clarity, realism, aesthetics), and **Prompt Following** (subject presence, attribute accuracy, action correctness, motion magnitude, camera movement) — plus an overall quality score. All reported scores are averages over Waver-Bench 1.0 (304 diverse prompts).
>
> ---
>
> **Q3: Higher-Resolution Comparison**
>
> We acknowledge that quantitative 720p comparisons would strengthen the paper. Due to computational constraints, we were unable to reproduce the full benchmark at 720p within the rebuttal period, but will include these results in the revised version.
>
> Importantly, we argue that higher resolution is a *favorable* setting for our method. At 720p the sequence length $N$ increases ${\sim}2.25\times$ over 480p. For a fixed sparsity ratio $s$, the number of retained tokens $(1{-}s) \times N$ grows proportionally, yet the critical visual structures (object boundaries, motion trajectories) do not scale at the same rate — higher-resolution sequences exhibit greater spatial redundancy. Thus, maintaining the same sparsity ratio at higher resolution becomes *easier* in terms of preserving essential attention structure.
>
> ---
>
> **Q4: Analysis of Table 2 — Veda vs. VSA on VBench**
>
> We respectfully argue that t
> he comparable VBench scores should not be interpreted as indicating similar perceptual quality. As detailed in our response to Reviewer SQWi (W1), VBench has well-documented limited discriminative power at this quality regime, as recent top-venue works consistently show that VBench lacks sufficient discriminative capacity when comparing strong modern models, making score differences unreliable.
>
> Our human evaluation reveals a substantially different picture. As shown in Figure 7, at 95% sparsity on Waver-1B, evaluators prefer Veda over VSA by 76% vs. 24% on Visual Quality and 39% vs. 16% on Overall Quality, despite near-identical VBench scores. We further conducted a **supplementary SBS evaluation** on **Wan2.1-1.3B** (64-prompt diverse subset, 90% sparsity): Veda is preferred **35.9%** vs. **17.2%** for VSA (46.9% ties), confirming the advantage generalizes across architectures beyond what VBench captures. Please refer to our response to Reviewer SQWi (W1) and Q2 above for further discussion on VBench limitations and human evaluation metrics.

---

> > ### Author Rebuttal · Reviewer_dUXi · 2026-04-02
> >
> > My concerns are all addressed.

---

### Official Review · Reviewer_LsXh · 2026-03-11

**Soundness:** 3
**Presentation:** 3
**Significance:** 2
**Originality:** 3
**Overall Recommendation:** 4
**Confidence:** 3

**Summary:**

This paper studies the quadratic self-attention bottleneck in video diffusion Transformers. The main claim is that quality degradation under high sparsity is caused less by the sparsity ratio itself and more by whether the sparse mask aligns well with the tile-wise structure of full attention. Based on this observation, the paper proposes Veda, which learns tile importance via full-attention distillation, uses TripPool features (mean/max/min), adopts head-aware tiling, and implements a custom tile-skipping kernel for efficient execution. Experiments on Waver and Wan show substantial speedups, with strong quality-efficiency trade-offs in many settings.

**Compliance With Llm Reviewing Policy:**

Affirmed.

**Key Questions For Authors:**

see above

**Limitations:**

yes

**Strengths And Weaknesses:**

strengths
1. Important problem with practical relevance. The paper addresses a real bottleneck in long-video diffusion models, and the contribution is not only algorithmic but also systems-oriented. The custom sparse kernel makes the speedup meaningful in wall-clock terms, not just in FLOPs.
2. Well-motivated method design. The empirical analysis is one of the strongest parts of the paper. In particular, the oracle-mask vs. pooling-mask comparison supports the claim that mask quality matters more than sparsity alone, and the tile-recall analysis provides a useful way to quantify alignment with full attention. The visualization of head- and timestep-specific patterns also gives reasonable support for head-aware tiling.
3. Strong empirical results overall. The reported end-to-end speedups are impressive, especially on larger/longer settings.

weaknesses
1. Baseline coverage is limited. The paper discusses both static-pattern and dynamic sparse attention methods, but the main comparisons are mostly against VSA and oracle masks.

---

> ### Author Rebuttal · Authors · 2026-03-31
>
> We sincerely thank Reviewer LsXh for championing our work and recognizing its "practical relevance," "well-motivated method design," and "impressive end-to-end speedups." We appreciate the opportunity to clarify our baseline selection rationale.
>
> ---
>
> **W1: Baseline Coverage Beyond VSA and Oracle Masks**
>
> We include the discussion of static-pattern sparsity (e.g., SVG) to provide a complete view of the evolution of the field. These methods represent an important earlier stage of exploration, where sparsity is imposed through predefined spatiotemporal templates. However, such designs are inherently limited in flexibility. As shown in our analysis in §3.2 (Figure 6), attention structures vary significantly across layers, heads, and timesteps. Under high sparsity ($\geq 90%$), fixed templates can lead to structural mismatches and result in the artifacts shown in Figure 3. This motivates the shift toward fully trainable dynamic methods.
>
> We agree that including direct comparisons with static-pattern methods would further improve the completeness of the evaluation. However, ensuring a fair comparison requires careful re-implementation and retraining under matched sparsity and hardware settings, which cannot be reliably completed within the rebuttal period.
>
> Prior work (e.g., VSA) has already demonstrated that dynamic sparse attention methods significantly outperform static-pattern approaches under comparable settings. Since our method further improves over VSA in the same regime, the relative performance trend with respect to static methods is expected to be consistent. Nevertheless, we agree that directly reporting these results would strengthen the paper.
>
> We will include additional comparisons with representative static-pattern methods (e.g., SVG) in the camera-ready version to provide a more comprehensive evaluation.

---

> > ### Author Rebuttal · Reviewer_LsXh · 2026-04-03
> >
> > My concerns are all addressed.

---

### Official Review · Reviewer_SQWi · 2026-03-15

**Soundness:** 3
**Presentation:** 4
**Significance:** 3
**Originality:** 3
**Overall Recommendation:** 4
**Confidence:** 3

**Summary:**

This paper studies sparse attention patterns in large video diffusion transformer models and proposes the Veda method to learn model's tile selection and select top-k tiles to accelerate attention procedure in video diffusion models due to the massive amount of tokens. This paper is particularly interesting because it addresses that the tile selection can be learned by a tiny proxy network and can be better than fully handcrafted as claimed. Experiments and ablation studies show that video diffusion transformers with Veda can delivery the near same performance with great speedups.

**Compliance With Llm Reviewing Policy:**

Affirmed.

**Final Justification:**

After reading the rebuttal by the author and comments by other reviewers, I keep my positive score.

**Key Questions For Authors:**

1. Could authors also perform the tile recall analysis on VSA attention sparse patterns to see to what values it can reach? This could enhance the comprehensive understanding of the comparison of attention tiles.

2. Could authors explain why Veda performs much better with Waver 1.0 than Wan 2.1, esp. with high sparse ratios? Might this because the score alignment is learned from datasets where Waver is trained on?

3. I am confused about the claimed 5.1x `end-to-end speed up` of Waver-T2V-12B in the abstract and the definition of  `end-to-end speed up` . Is attention layer only included or the full video generation pipeline included? After all Table 2 speed up (max 2.6) cannot persuade me there is 5.1x speed up of 12B 720P model.

**Limitations:**

Yes. The limitations and potential negative societal impact is included.

**Strengths And Weaknesses:**

Strengths:

1. The paper is well written and clearly describe the technical details with figures. One can fully reproduce the code of the main idea of this paper.

1. The most interesting and appealing point of this paper, in my opinion, is the analysis of why current attention sparsifying methods in video diffusion transformers fails due to the incapable and flawed-designed tiles. The authors give detailed step-by-step analysis on that and can potentially invoke research interests to generally broad audience on sparse attention tile patterns in video diffusion transformers. This type of analysis should be encouraged in papers and well motivates the introduction of Veda.

1. The authors also perform detailed analysis of the proposed Veda acceleration, including layer- and component-wise latency analysis, and key ablation designs.

Weaknesses:

1. The most weak part of this paper, in my opinion, is the final result, especially compared to the recent rivals, VSA. First, in Table 2, the evaluation score of Veda (90%) closely match the VSA with few seconds reduced. Honestly, this is not significant improvements since Veda requires training on probably a lot of videos to learn attention tile patterns after all. When increasing the sparse ratio to 95%, the evaluation score drops significantly for Wan2.1 while the latency gains might become negligible. These results, in turn, recall why the attention tile patterns should be learned given the handcrafted one is good enough.

---

> ### Author Rebuttal · Authors · 2026-03-31
>
> We sincerely thank the reviewer for recognizing our analysis of attention sparsification failures, the detailed latency profiling, and the reproducibility of our approach. Below we address each concern.
>
> ---
>
> **W1(1): About Performance Gain over VSA**
>
> We respectfully argue that close VBench scores do not reflect the true quality gap, for two reasons:
>
> VBench has limited discriminative power at this quality regime. Recent top-venue work has documented this limitation: Dual-IPO (Yang et al., 2026) observed that as human-aligned rewards rose, VBench scores declined; EquiVDM (Liu & Vahdat, 2025) demonstrated VBench's static bias, as frozen videos score near-perfectly on Background Consistency while physically correct motion is penalized. When two strong methods both operate beyond VBench's discriminative capacity, score differences become unreliable.
>
> **Human evaluation reveals a significant quality gap.** As shown in Figure 7, at 95% sparsity on Waver-1B, humans overwhelmingly prefer Veda: **76% vs. 24%** on Visual Quality and **39% vs. 16%** on Overall Quality, despite near-identical VBench scores (e.g., Image Quality 0.698 vs. 0.692). Please refer to Q2 in our response to Reviewer dUXi for more detailed discussions on human evaluation metrics.
>
> To strengthen this, we conducted a **supplementary human evaluation on Wan2.1-1.3B** (64-prompt diverse subset, 90% sparsity). Results corroborate our findings: Veda preferred **35.9% vs. 17.2%** for VSA (46.9% ties), confirming the advantage generalizes across architectures beyond what VBench captures.
>
> ---
>
> **W1(2): About Training Requirements and Why Learned Patterns Matter?**
>
> **VSA is not training-free.** Both VSA and Veda are trained on the same number of videos with identical learning rates and total steps (Section 5, Implementation Details). We explicitly keep the total training budget fixed to ensure a fair comparison.
>
> Moreover, Veda's two-stage strategy is *more efficient*: Stage 1 freezes the backbone and trains only the lightweight score projector; Stage 2 fine-tunes from this initialization. VSA instead requires a handcrafted slow warmup to stabilize its sparse patterns.
>
> **On why learned patterns outperform handcrafted ones:** All compared methods (Oracle Mask, VSA, Veda) generate *dynamic, content-adaptive* mask, none is static. The distinction is *how* tile importance is determined. As shown in Section 3.2 (Observation 2), mask quality dominates generation quality far more than sparsity ratio itself. The Oracle Mask validates this but is impractical (requires computing full attention first, *slower* than dense inference: 94.6s vs. 58.5s on Wan2.1-1.3B). Veda approximates this oracle ranking via a lightweight estimator at negligible overhead.
>
> ---
>
> **Q1: Tile Recall Analysis on VSA**
>
> We thank the reviewer for the suggestion. However, tile recall analysis on VSA is not directly meaningful due to a fundamental mismatch in objectives.
>
> Tile recall measures alignment with a full-attention oracle mask. Our method is explicitly trained to match this oracle via distillation. In contrast, VSA uses average-pooled representations with Top-K selection without any supervision to align with full attention, making its tile recall not a meaningful indicator. Notably, the "Average Pooling Mask" in Fig. 5 (34.2% tile recall) closely corresponds to VSA's core mechanism, already serving as a proxy.
>
> ---
>
> **Q2: Why Better on Waver 1.0 than Wan 2.1?**
>
> **The perceived gap is largely a VBench artifact.** As discussed in W1, VBench has limitations in discriminating strong models—saturating at this quality regime and exhibiting systematic biases. The VBench score differences between Veda on Waver vs. Wan2.1 therefore reflect varying metric sensitivities across architectures, not a genuine quality disparity. Our supplementary human evaluation on Wan2.1 (35.9% vs. 17.2% preference at 90% sparsity) confirms that Veda's perceptual advantage holds consistently across both models.
>
> ---
>
> **Q3: Clarification on 5.1× End-to-End Speedup**
>
> We thank the reviewer for raising this point. The 5.1× refers to **wall-clock end-to-end speedup of the full generation pipeline**, encompassing all transformer layers, sparse mask prediction, and VAE decoding.
>
> The apparent discrepancy with Table 2 is attributable to the difference in **resolution and sequence length**. Table 2 evaluates at **480P / 81 frames** with moderate token count $N=34{,}020$, where self-attention constitutes a comparatively smaller fraction of total pipeline latency. The 5.1× speedup is instead measured on Waver-T2V-12B at **720P / 241 frames** ($N = 245{,}760$).
>
> This difference is a direct consequence of the $O(N^2)$ complexity of self-attention versus $O(N)$ for all remaining components. As $N$ increases with higher resolution and longer duration, **the proportion of total compute attributable to attention grows substantially**, causing sparse attention savings to yield correspondingly larger end-to-end gains.

---

> > ### Author Rebuttal · Reviewer_SQWi · 2026-04-04
> >
> > Thanks for the detailed response. I will keep my positive score.

---

### Decision · Program_Chairs · 2026-04-30

**Decision:**

Accept (regular)

**Comment:**

All four reviewers unanimously gave a final rating of Weak Accept (4). The authors propose a highly practical distilled sparse attention framework that accelerates large video diffusion models by formulating tile selection as an explicit reconstruction problem. Throughout the review process, the reviewers praised the paper's insightful empirical analysis, which proves that sparse mask alignment with full attention geometry matters more than the sparsity ratio itself, as well as the impressive 5.1x wall-clock speedup achieved via a custom hardware-efficient kernel. While initial critiques noted limited baseline comparisons and a reliance on the potentially saturated VBench metric, the authors provided a comprehensive rebuttal that fully resolved these issues. By providing supplementary human evaluation results and clarifying the architectural portability of the tile-skipping mechanism beyond NVIDIA Hopper, the authors addressed all reviewer concerns. Ultimately, this is a well-motivated and impactful contribution that effectively addresses the critical quadratic compute bottleneck in long-video generation.  I confidently recommend this paper for acceptance.